# Nanostructured Thick Electrode Strategies toward Enhanced Electrode–Electrolyte Interfaces

**DOI:** 10.3390/ma16093439

**Published:** 2023-04-28

**Authors:** Anukriti Pokhriyal, Rosa M. González-Gil, Leandro N. Bengoa, Pedro Gómez-Romero

**Affiliations:** Novel Energy-Oriented Materials Group at Catalan Institute of Nanoscience and Nanotechnology (ICN2) CSIC and BIST, Campus UAB, Bellaterra, 08193 Barcelona, Spain; anukriti.pokhriyal@icn2.cat (A.P.); rosamaria.gonzalez@icn2.cat (R.M.G.-G.);

**Keywords:** nanostructure, thick electrode, electrolyte–electrode interface, energy storage, energy density, supercapacitor

## Abstract

This article addresses the issue of bulk electrode design and the factors limiting the performance of thick electrodes. Indeed, one of the challenges for achieving improved performance in electrochemical energy storage devices (batteries or supercapacitors) is the maximization of the ratio between active and non-active components while maintaining ionic and electronic conductivity of the assembly. In this study, we developed and compared supercapacitor thick electrodes using commercially available carbons and utilising conventional, easily scalable methods such as spray coating and freeze-casting. We also compared different binders and conductive carbons to develop thick electrodes and analysed factors that determine the performance of such thick electrodes, such as porosity and tortuosity. The spray-coated electrodes showed high areal capacitances of 1428 mF cm^−2^ at 0.3 mm thickness and 2459 F cm^−2^ at 0.6 mm thickness.

## 1. Introduction

We are witnessing an accelerated transition from fossil fuels to renewable energies, and energy storage in all its forms and facets is being called on to play a central role in this transition. Among all possible technologies, electrochemical energy storage (EES) systems provide great versatility, from large stationary storage to electric vehicle (EV) batteries to minuscule flat and flexible supercapacitors. However, there are many challenges still to be met for these solutions to be truly optimal and allow for the massive deployment of EES systems.

When it comes to the development of better EES, emphasis is generally on electrode composition, especially on the nature of the active phase, its microstructure and reactivity. When it comes to electrode engineering, electrodes are invariably fabricated as thin solid coatings on thicker current collectors, both for rechargeable batteries and for supercapacitors. It is evident that this is not an optimal design. However, the implementation of thicker electrodes has traditionally run into dead ends. On one hand, a mere increase in the thickness of electrodes using conventional formulations systematically leads to poorer performance. On the other hand, better performance is only achieved by designing ultra-sophisticated nanostructures that are difficult to scale up and implement. Therefore, it is necessary and urgent to address the need for thicker electrodes through fundamental studies that could help understand the factors hampering their performance.

Carbon-based supercapacitor electrodes generally have a thickness of a few microns, and the ratio of active components (energy storing) to non-active components (binder, current collector, separator, casing, etc.) is very low. In order to reduce costs and make compact energy storage devices with high power densities [1,2,3], it is necessary to increase the active material loading through the development of thick or high mass loading electrodes and subsequently lower the quantity of non-active components. The industry standard for the mass loading of a thick electrode is generally >10 mg cm^−2^; however, such thick electrodes suffer from low capacitance mainly due to poor kinetics, reduced structural stability and high tortuosity [4]. Furthermore, delamination of the electrode from the current collector constitutes a possible complication leading to higher internal cell resistance [1].

Many approaches have been followed to develop thick electrodes that perform as well as conventional thin electrodes. Common methods to improve charge transfer include the development of hierarchical porous composites though pyrolysis [5,6,7,8], design of current collectors [9,10,11,12], addition of conductive materials [13,14,15,16], and modification of binders [17,18,19,20].

The introduction of carbon additives to activated carbon (AC) has long been studied to improve conductivity in supercapacitors and develop electrodes with higher areal and volumetric capacitance. AC has high porosity and thus good capacitance; however, it has low conductivity [21]. Commercial conductive materials such as carbon blacks (carbon super-P (CSP) or C65), nanofibers (NFs) and carbon nanotubes (CNTs) are added to improve electronic conductivity [22,23,24]. For example, a 500 µm thick CNT-based supercapacitor with bottom-up gel-filled electrolyte showed an areal capacitance of 2662 mF cm^−2^ at a scan rate of 2 mV s^−1^ and 2038 mF cm^−2^ at 10 mV s^−1^ [25]. In another study, a millimetre-thick supercapacitor electrode using poly(3,4-ethylenedioxythiophene) as the active material showed a specific capacitance of 6596 mF cm^−3^ at 25 mV s^−1^ and demonstrated high mechanical strength. In another example, a supercapacitor developed using single PANI-incorporated ultra-thick carbon nanotube fibre exhibited a high specific capacitance of 335 F g^−1^ and volumetric capacitance of 523.3 F cm^−3^ at 1 A g^−1^ [26]. A free-standing flexible electrode made of Co-UTBiOBr and CNTs with mass loading of 15 mg cm^−2^ showed rate capability of 61 mAh g^−1^ at 1 A g^−1^ after 3000 cycles [27]. 

Improving ion transfer throughout the bulk of the thick electrode is another approach to improve thick electrode performance. Reducing the tortuosity of the electrode by pore engineering has provided some promising results [28,29], and various techniques can be used to optimize the nano/micro pore structure of the electrode to improve ion transfer [2,30,31]. For example, electrolyte saturation can be improved through lyophilization or freeze-casting to create nanochannels and reduce tortuosity [4,32,33], and a ∼1 mm thick cellulose-based asymmetric supercapacitor with a RuO_2_/cellulose carbon aerogel anode achieved a high areal capacitance of 4284 mF cm^−2^ at 2 mA cm^−2^ [34]. In another study, a two-step tape-casting/freeze-casting method resulted in the formation of low tortuosity graphene electrodes that demonstrated enhanced ionic transfer [30].

Apart from introducing carbon additives to improve conductivity and creating porosity to improve ionic conductivity, different coating and electrode formulation methods may also improve the performance of thick electrodes. The thick electrodes made using conventional knife casting methods have shown poor adhesion between carbon particles and higher series resistance with cycling, leading to reduced capacitance [35]. Moreover, thicker slurries may not allow good dispersion of carbon particles and binder. To overcome this, fine inks can be developed and sprayed on collectors in a controlled manner to allow better dispersion and good contact between carbon particles [24,36] as well as to control the thickness of the electrode in an efficient way [37,38].

Despite these various methods intended to solve the issues related to thick electrodes, there are few or no reports on thick electrodes with good rate capability that could be prepared by a scalable method. In this article, a multi-faceted approach was followed to develop high mass loading multilayer electrodes (up to ∼0.6 mm). This approach included testing two different coating methods, as well as different conductive additives and binders. Simple scalable methods such as spray coating and freeze casting were tested, and the effects of carbon additives such as CSP and CNTs as well as binders such as cellulose and PVDF-HFP were analysed and compared. This study also sought to shed some light on the barriers that hinder the performance of thick electrodes, thus helping to define design strategies.

## 2. Materials and Methods

### 2.1. Materials

All raw materials were commercially available and were used without additional purification or modification. Activated carbon for supercapacitor applications (YP50F) was purchased from Kuraray (bulk density of 0.3 g mL^−1^ and surface area of 1692 m^2^ g^−1^) and Carbon black Super P (≥99%) was purchased from Alfa Aesar (density of 160 ± 20 kg m^−3^). The remaining materials, Multiwalled Carbon Nanotubes (CNTs, 6–9 nm × 5 µm), 1-Methyl-2-pyrrolidone (NMP), Poly(vinylidene fluoride-co-hexafluoropropylene) (PVDF-HFP) and Carboxymethylcellulose (CMC) 90,000 Mw, were purchased from Sigma Aldrich, Madrid (Spain). Aluminium foil of 0.05 mm thickness was used as current collector (Goodfellow). The fabricated electrodes were assembled in CR2032 coin cells from MTI by using Whatman grade 4 cellulose paper separator and 1M tetraethylammonium tetrafluoroborate (TEABF_4_) in acetonitrile (all purchased from Sigma Aldrich) as electrolyte. 

### 2.2. Active Slurry for Electrode Preparation

Different compositions of the active ink were prepared for comparisons. It has been observed that a conductive carbon proportion of 10–15% shows the best performance [39] and hence, a 10% conductive carbon percentage was chosen for our electrodes of all thicknesses and compositions. CMC was selected as binder due to its environmental friendliness and flexibility [40], while PVDF-HFP was also tested because it has previously been used in the development of gel-electrolyte flexible supercapacitors [41]. Finally, it is well known that binders can block AC pores, causing a reduction in the capacitance of electrodes. Thus, binder mass was kept as low as possible while maintaining structural stability of the coatings for both CMC and PVDF-HFP binders. AC was combined with either CSP or CNTs as conductive additives, and CMC as the binder material. An AC: conductive additive: CMC ratio of 85:10:5 in de-ionized water and stirred for 12 h until a spreadable slurry was obtained. The compositions of AC, CSP and PVDF-HFP (80:10:10 and 75:10:15) was prepared by stirring with NMP for 12 h (Appendix A).

### 2.3. Coating Methods

Spray coating and freeze casting (or lyophilisation) were chosen as the two scalable methods for making thick electrodes with a controllable structure. The knife casting method was not considered due to the formation of cracks in the thicker coatings upon drying. Figure 1 depicts a schematic of the electrode preparation processes by spray coating and freeze casting.

#### 2.3.1. Spray Coating

For the AC (CSP/CNT) CMC electrodes, the aluminium current collector was placed on a heating plate with a constant temperature maintained at 60 °C. The slurry was loaded into a spray gun, and coatings were produced by controlling the number of sprays (4, 15, 35 and 50, resulting in electrode mass between 1–2, 6–8, 10–14, and 17–24 mg, respectively). The deposit was allowed to dry for 30 s between each coating to prevent solvent concentration and subsequent cracking. The coated film was then pressed at 3 Mt in a mechanical press and dried overnight at 100 °C. This helped maintain consistency in mass and volume between different coatings and samples (Appendix A).

In the case of PVDF-HFP as the binder, only CSP was used as the conductive material. AC, CSP and PVDF-HFP were dissolved in NMP in ratios of 80:10:10 and 75:10:15. The process was the same as above except for a few differences. For example, the coatings were produced at a constant heating temperature of 75 °C, and the slurry was sprayed 3 and 27 times to obtain coatings of different thicknesses and mass loadings (Appendix A). 

To test the limits of the spray coating process, even thicker electrodes (0.6 mm) with AC:CNT:CMC (85:10:5) as well as AC:CSP:PVDF-HFP (80:10:10) were developed with mass of 48 mg and 46 mg, respectively. The electrode names have a number as a prefix to differentiate different coatings on the basis of mass. For example, electrodes between 1 and 2 mg have a prefix of 1, while those between 6 and 8 mg have a prefix of 2, and so on.

#### 2.3.2. Freeze Casting

For freeze casting, AC, CSP and CMC were mixed in a ratio of 85:10:5 in water. The slurry was coated onto the current collector using a casting knife and then frozen in a container with liquid nitrogen. To obtain a thicker coating, the slurry was subsequently coated and frozen three times. The coatings were then placed in a lyophiliser for 24 h to completely remove the solvent. The electrodes were not pressed except for the cell “3B_LYO_5%CMC”.

### 2.4. Cell Assembly

Symmetrical CR2032 coin cells were assembled for each type of thick electrode prepared in this work. After drying in a vacuum oven overnight, electrodes were cut to Φ = 14 mm size and then pressed. The thickness was measured with a micrometre, and the electrodes were further dried at 100 degrees for 6 h. They were transferred to an argon-filled glovebox (Jacomex GP with O_2_ < 5 ppm and H_2_O < 5 ppm), and electrodes with similar mass were placed with a cellulose separator (Φ = 16 mm) in the coin cell with 1M TEABF_4_ in acetonitrile (ACN) as the electrolyte for electrochemical characterization.

### 2.5. SEM Analysis

Electrode surface morphology was analysed by SEM (FEI Quanta 650 FEG) without metallization. Cross-sections were prepared by blade cutting and placed onto aluminium stubs at an angle of 70° to analyse the electrode structure of the different casting methods and characterize their thickness. Images were taken at 5 kV at different magnifications for each sample.

## 3. Results and Discussion

### 3.1. Morphology Characterization

Figure 2 and Figure 3 show the cross-sections of thin electrodes (left column) and thick electrodes (right column). For spray-coated electrodes, comparisons were made between electrodes having thicknesses in the range of 20–30 µm (suffix 1) and 317–333 µm (suffix 4). For the freeze-casted electrodes, the thinnest 271 µm (2_LYO_5%CMC) and thickest 241 µm (3A_LYO_5%CMC) electrodes where selected. 

Even surfaces and good contacts with the current collectors could be observed for all electrodes. As the thickness increased, some thick electrodes showed minor unevenness on the surface due to the hand-coating process. Nevertheless, the structural integrity of the electrodes was maintained with the increase in thickness. A compact layering of carbon could also be seen for all spray-coated electrodes. 

In the case of the thin freeze-casted electrodes (Figure 2e), a distinct macro pore structure was not observed. The electrode structure resembled the thin electrodes made with the spray-casting method without showing any pattern. However, in the case of thicker freeze-casting electrodes, a characteristic macro pore structure could be observed, which was inherent to the freeze-drying process. This macrostructure was more open and distributed like an accordion, in which nano-channels may allow for better electrolyte percolation. Nevertheless, the macro pores were unevenly distributed and randomly oriented, creating dead-end pores that could be ascribed to the casting of multiple layers at inconsistent temperatures.

### 3.2. Porosity Analysis

A porous electrode allows better electrode percolation throughout the bulk and, therefore, better utilisation of active material [42]. Therefore, having an ideal pore structure becomes an important consideration for thick electrodes. The porosity (P) of an electrode is given by Equation (1):(1)$$P=\left[1-\frac{\rho_{g}}{\rho_{t}}\right]\cdot 100$$
where $\rho_{g}$ is the geometrical density of the electrode (g cm^−3^) based on the thickness and loading of the electrode, and $\rho_{t}$ is the true electrode density (g cm^−3^) based on the density of the slurry.
(2)$$\rho_{g}=\frac{mass\;loading\;of\;electrode\;({g\;cm}^{-2})}{d}$$
and
(3)$$\frac{1}{\rho_{t}}=\sum \frac{\omega_{i}}{\rho_{i}}$$

d is thickness of electrode without the aluminium foil in cm. $\rho_{t}$ is the true density of the materials, $\omega_{i}$ is the mass fraction of each material of the electrode, and $\rho_{i}$ is the density of each component.

All electrodes had porosity values of >80% (Figure 4). For spray coatings, the thin electrodes had a greater porosity (≥85%), which decreased with the increase in thickness (Table 1). On the other hand, electrodes produced with the freeze-casting method showed very high porosity even at higher thicknesses, due to their more open structure. The thickest electrode produced by freeze casting (∼600 µm) had the highest porosity of all (96%). When the thick electrode was compressed and the macro-pore structure was destroyed, the porosity was reduced to 85%. However, as will be shown below (Figure 4), the increase in areal capacitance showed that the porosity was sufficient to allow adequate electrolyte infiltration and, therefore, ion transfer. Gravimetric and volumetric capacitance variance with porosity and thickness were also calculated and can be seen in Appendix A. Taking into account only the gravimetric capacitance, thinner electrodes ( 300 µm) had the better performance, and porosity did not play a significant role. However, when volumetric capacitance was observed, an independence from the thickness appeared, and the most “compact” electrodes performed better. It was thus observed that the porosity values between 80–90% were desirable to have good contact between carbon grains and increase the capacitance of the thick electrodes.

### 3.3. EIS and Tortuosity Calculation

Electrochemical impedance spectroscopy (EIS) can be used for detailed information on the kinetics of the charge–discharge process as well as the pore structure of electrodes [43,44]. The EIS experiment was conducted over a frequency range from 500 kHz to 50 MHz with a 10 mV perturbation amplitude using a BioLogic VMP3 potentiostat controlled by the EC-Lab software. The equivalent circuit was obtained by fitting the impedance spectra using a simplified transmission-line model with constant-phase elements (TLM-Q, Ma_3_) that neglected the charge transfer across the electrode/electrolyte interface (i.e., blocking conditions) [45]. Figure 5a–c shows a semicircle followed by a transmission-line behaviour for all of the samples. 

The high-frequency semicircle has been previously observed in the analysis of both thick and thin electrodes and has been ascribed to different phenomena. For example, it has been assigned to poor contact between the electrode and the current collector [45,46], to the electrolyte resistance within the pores of the electrode [44], or even to redox processes of heteroatoms present in the carbon materials [46]. The smaller semicircle for thick electrodes could represent lower interface charge loss as compared to the thin electrodes [47,48]. Further, Figure 5a,b show that the resistances of 1_CSP_5%CMC were smaller than those of 1_CNT_5%CMC whereas the resistances of 4_CSP_5%CMC were larger than those of 4_CNT_5%CMC. This anomaly could be due to the fact that CSP was observed to provide slightly better charge transfer between AC particles than CNTs, leading to lower resistance for thin electrodes. However, CNTs provided better mechanical strength (and, therefore, slightly better adhesion to the current collector), which was a more important factor in the case of thick electrodes (4_CSP_5%CMC and 4_CNT_5%CMC). Thus, CNT-based thicker electrodes had a lower resistance. 

Regardless of the cause leading to this electrochemical response, it has been proved that, as long as the transmission-line (TL) behaviour can be observed, the ionic resistance and so the tortuosity of electrodes can be determined [45]. Therefore, the circuit proposed is still valid to estimate the electrode tortuosity to correlate this parameter with the electrochemical performance. Together R1, R2 and Q2 represent the internal resistance of the electrode. Ma3 is the modified restricted diffusion element that represents ion diffusion within the electrode together with the capacitive charging of the surfaces, modelled with constant-phase elements to account for non-ideal behaviour that arises from geometric effects. The slope of the line at intermediate frequencies differed for different thicknesses, indicating a longer ion diffusion process in the thicker electrodes, as could be expected [44]. The model also encompasses the diffusion resistance Rd3, which gives the total ion transport resistance or R_ion_. The inductor L1 represents a negligible inductance in the electrode [49]. The flow of electrolyte through the pores of the electrodes can be determined by the tortuosity (τ), which can be calculated from Equation (4) [30]:(4)$$\tau =\frac{R_{ion}\cdot A\cdot \kappa \cdot P}{2d}$$
where A, d, and P are surface area, thickness, and porosity of the electrode, respectively, while κ is the conductivity of the electrolyte (in our case, 0.05614 S cm^−1^).

The tortuosity values for the electrodes are reported in Table 1 and Appendix A. The thin electrodes with lower binder percentages reported lower tortuosity values (in the case of using CMC as binder); however, tortuosity nearly doubled each time as binder percentage increased from 5% to 10% to 15%, being indistinct to the binder nature. On the other hand, as thickness increased, the tortuosity of the electrodes decreased and, in some cases, the values observed were below 1 (tortuosity for through-pores). This could be possible due to the presence of dead-end pores, which would contribute to an increase in the surface area causing this decrease in tortuosity [50]. The study by Nguyen et al. points out that there is no need for ions to percolate all the way to the current collector as long as they reach the active surface of the electrode. Therefore, dead-end pores might not only lead to an increase in the active area but also provide more easily accessible paths for ions. According to this study, the latter translates into a reduction in the tortuosity value, which could even take values lower than one (more accessible than straight pores). It is worth mentioning that the layer-by-layer deposition used in this work may contribute to the formation of these pores. Likewise, dead-end pores may be caused due to the addition of conductive carbon as well as the binder. The results show that the electrodes with CSP had lower tortuosity values compared to the other spray-coated electrodes. Further, freeze-casted electrodes also had a very low tortuosity value due to the presence of micropores and more open structure. However, as can be seen from Figure 2e and Figure 3e, these pores were not aligned or well-ordered, which could have resulted in dead-end pores mentioned earlier and hence tortuosity values < 1.

### 3.4. Electrochemical Characterization

Electrochemical tests were performed with a VMP3 multichannel potentiostat (Bio-Logic) using the EC-Lab software. Cyclic polarization (CP) was performed for symmetrical two-electrode cells between 0 < V < 2.7, at different scan rates (2 < mV s^−1^ < 200) to evaluate the capacitive behaviour and calculate the potential window, stability, and cell capacitance (F) of the supercapacitors and the specific (F g^−1^), areal (F cm^−2^), and volumetric (F cm^−3^) capacitances of each electrode. Galvanostatic charge–discharge (GCD) was also performed to characterize the charging behaviour and obtain another estimate of the gravimetric capacitance.

The cell capacitance (C_cell_) of a symmetrical capacitor can be calculated from cyclic polarisation (CP) as:(5)$$C_{cell}=\frac{1}{2}\left[\frac{∆Q}{∆V}\right]=\frac{\int I\left(V\right)\cdot dV}{2\cdot \upsilon \cdot ∆V}$$

In the case of symmetrical supercapacitors, $\int I\left(V\right)\cdot dV$ is the total integral of the area under the CP (A_CP_) (Q_charge_ + Q_discharge_), υ is the scan rate of the CP, and ΔV is the working potential window. The capacitance of an electrode (C_e_) is twice that of the cell (C_cell_); therefore,
(6)$$C_{e}=2C_{cell}$$

The electrode volumetric capacitance (C_v_), areal capacitance (C_a_) and gravimetric capacitance (C_g_) can be calculated as:(7)$$C_{v}=\frac{C_{e}}{v}$$
(8)$$C_{a}=\frac{C_{e}}{A}$$
(9)$$C_{g}=\frac{C_{e}}{m}$$
where v is the volume of the electrode without the current collector (cm^3^), A is the area (cm^2^), and m is the mass of the electrode (g) considering only the active material. 

Finally, gravimetric energy density (Wh kg^−1^) and gravimetric power density (W kg^−1^) of the device were calculated at 1 A g^−1^ from the equations below, where t is the discharge time.
(10)$$E_{g\_cell}=\frac{1}{8}\left[\frac{C_{g}{∆V}^{2}}{3.6}\right]$$
(11)$$P_{g\_cell}=\frac{E_{g\_cell}}{t}$$

#### 3.4.1. Capacitance

Figure 6 shows the CP curves of different electrodes with increasing thicknesses at a scan rate of 10 mV s^−1^. The graphs show that as the thickness of electrodes increases, the area under the graph also increases, signifying and increase in areal capacitance. Comparing methods, it is noted that the freeze-casted electrodes had significantly lower capacitance compared to the spray-coated electrodes; however, the freeze-casted and pressed electrode (3B_LYO_5%CMC) showed a better areal capacitance and rate capability than its unpressed counterpart (3A_LYO_5%CMC), revealing that while the macro pore structure allows better electrolyte percolation, it reduces the contact between carbon particles, leading to poor electron transfer.

On the other hand, the effect of scan rate on the areal capacitance of different samples is shown by Figure 7. An expected trend of reduction in capacitance with increase in scan rate was observed for all materials. The thick electrodes demonstrated high areal capacitances at high scan rates. However, the spray-coated thick electrodes’ capacitances rapidly declined after 20 mV s^−1^. This was not seen in the case of thin electrodes, which retained their areal capacitances even at high scan rates.

The thickest electrode produced using CNTs as carbon material and 5%wt CMC as binder (5_CNT_5%CMC, ∼600 µm and 31 mg cm^−2^) showed the best aerial capacitance of 2459 F cm^−2^. The only electrode with a similar mass loading and thickness (5_CSP_10%PVDF-HFP) had a slightly lower capacitance of 2261 F cm^−2^ at 10 mVs^−1^, using 10% wt of PVDF-HFP as binder. However, these electrodes did not perform well at high scan rates. The variation in aerial capacitance (mF cm^−2^) with changes in mass loading (mg) and thickness (µm) of the electrodes with different materials can also be visualized through Figure 8. Both gravimetric and volumetric capacitance dependences are also depicted in Appendix A from Supporting Information. As in the comparison of the porosity, volumetric capacitance is independent of the thickness, being comparable to the performance of thick electrodes and the ones between 100–200 µm, and gravimetric capacitance is better for thin electrodes. 

#### 3.4.2. Specific Energy and Power

Figure 9 shows the specific energy versus power for selected electrodes at a current density of 1 A g^−1^. Overall, as before, spray-coated electrodes performed better than freeze-casted electrodes. Depending upon the thickness, there was a large variation in the specific power of the electrodes. The thickest electrodes outperformed the thin electrodes, as they had the highest specific power, at 942 W kg^−1^. 

It was observed that the specific energy values of devices made with different thicknesses, carbon additives and binders were largely comparable. However, the thin electrode with CSP as the additive and 15% PVDF-HFP as a binder had slightly better, but comparable, specific energy (27 Wh kg^−1^) than the thick electrodes with CMC as the binder (26 Wh kg^−1^).

#### 3.4.3. Impact of Carbon Additives, Binders and Thickness

The role of binder in the capacitance of electrodes can also be assessed from Figure 7. CNT-based electrodes had slightly better areal and volumetric capacitance than CSP-based electrodes, as they are known to disperse better with CMC. Interestingly, while the overall areal capacitance of CNT electrodes was higher than that of CSP, the slope of capacitance vs. scan rate was also higher. That is, CSP-based electrodes demonstrated a lower reduction in performance as the scan rate increased. Moreover, despite the better bonding and better mechanical strength of CNT electrodes, CSP-based electrodes of similar thickness had better gravimetric capacitance (Appendix A). Therefore, carbon-to-carbon contact in thick electrodes is better with CSP, although the thickness cannot be increased over a certain limit due to the development of cracks, unlike CNT-based electrodes. 

Overall, thick electrodes with 5% CMC as a binder had superior performance, considering gravimetric capacitance (Appendix A), than those with 10% or 15% PVDF-HFP. Nevertheless, the PVDF-HFP-based thick electrodes were slightly more flexible than CMC electrodes. Further, it can be noticed that the increase in the amount of PVDF-HFP binder (from 10% to 15%) had a minor impact on the capacitance of the electrode. Thick electrodes produced using 15% PVDF-HFP had slightly better performance at high scan rates. This could be attributed to their better adherence to the current collector. It has been well established that using large amounts of binder can block carbon pores and reduce capacitance, which is also supported by the tortuosity analysis [50]. However, in this case, the improvement in gravimetric capacitance of the thick electrodes could be explained by the possibly better adherence with the current collector.

#### 3.4.4. Discussion

The use of carboxymethylcellulose (CMC) as binder with CNTs as conductive additives led to electrodes with good mechanical strength but, not especially flexible, whereas thick electrodes developed using PVDF-HFP copolymer as the binder not only had high mechanical strength but were also flexible. Nevertheless, these latter electrodes required a higher percentage of the binder, which increased the percentage of non-active components in the supercapacitor. Moreover, since the areal capacitance of the CMC and PVDF-HFP thick electrodes was similar, CMC with CNTs was judged to be a better option to develop thick electrodes, since CMC is environmentally friendly, soluble in water, and is required in lower quantities.

Thick electrodes developed using these methods featured increased areal capacitances with increases in mass loading and thickness. However, this was not observed in the case of gravimetric capacitances at high scan rates. While the thick electrodes had high gravimetric capacitance at low scan rates, at higher scan rates, the thin electrodes not only performed much better, but also did not suffer loss in performance. Nevertheless, comparing the areal capacitance alone (as is common practice for thick electrodes) can be misleading. Volumetric capacitance could give a more realistic picture of the material performance, as it would consider the bulk of the electrode and, in a more practical sense, it would be more useful for device design. Nevertheless, it ignores the utilisation of active material, which can be determined with the analysis of gravimetric capacitance. Thus, a comprehensive analysis of gravimetric, areal and volumetric capacitance should be carried out for each specific device to be developed to assess the performance of thick electrodes and ensure compact cells, as well as the optimal utilisation of the active material. In line with this conclusion, carbon-based supercapacitor electrodes with mass loading between 5 and 8 mg cm^−2^ and with thickness between 100 and 200 µm had the best performance at high scan rates.

Porosity estimates and tortuosity analyses from impedance measurements were also performed to determine the pore structure of the electrodes. It was observed that dead-end pores in thick electrodes could be causing anomalously low tortuosity values, and thus, other methods such as electron tomography and X-ray absorption spectroscopy could be used in the future to further reveal the pore structure. Furthermore, freeze-casted electrodes had high porosity, but they did not demonstrate high capacitances. Therefore, a higher degree of porosity in the material does not necessarily ensure better performance, as there needs to be a balance between porosity and the contact between carbon particles within the bulk of the electrode material. Thus, an ordered pore structure with good contact between carbon particles, as depicted in Figure 10d, would ideally have better performance across all metrics studied in this paper.

This is the key issue in the development of thick electrodes. Traditionally, the good balance between electrolyte impregnation of an electronically conductive carbon network has been “solved” by fabricating thin, single-layer electrodes (Figure 10a). However, this obviously represents renouncing the great potential of large, “bulk” electrodes to provide much higher capacity and energy density per device. This ideal goal is represented by the idealized drawing in Figure 10d, with porosity fully extending throughout multilayer-thick electrodes. In this work, we were able to highlight that the later goal is possible (though by no means too easy), and at the same time, we witnessed the many barriers that can be found to reach it. For instance, the fabrication of freeze-casted electrodes, as carried out by us and reported here, led to multilayer electrodes with large but dead-end porosity (as shown in Figure 10c) and poor performance. On the other hand, multilayer-sprayed thick electrodes led to much-improved areal capacitance with respect to thin electrodes. The fact that this improvement is not realized when normalizing by mass means that there is still room for the improvement of our thick electrodes. Finally, when the normalization of capacitance is conducted per unit of volume, an interesting situation arises: thick electrodes (multi-spray method) feature higher volumetric capacitances than thin electrodes, but only at low scan rates. This tells us that presently our thick electrodes conform to an intermediate situation such as that depicted in Figure 10b, where electrolyte impregnation of the bulk solid electrode can be good but where other factors impede ionic diffusion at high rates. Determining and solving these impeding hurdles must be our next collective goal.

## 4. Conclusions

Thick electrodes (up to 0.6 mm) were developed using commercial carbons and additives and low-cost, easily scalable methods, namely, multi-spraying and freeze-casting methods. The multiple spray coating method was developed and successfully tested to obtain electrodes with high mass loadings (31 mg cm^−2^). Different binders and their compositions were also tested to ensure coatings with high mechanical strength. Freeze-casting was applied to pre-formed electrodes to test the effect of water sublimation in the formation of macropores and the corresponding effect on performance. This latter method led to large macropores but much lower mass loading. The electrodes developed using these methods showed very high areal capacitances. For instance, a 0.6 mm multilayer spray-coated electrode with CNTs featured an areal capacitance of 2459 F cm^−2^ at 10 mV s^−1^, and the corresponding symmetrical devices showed good specific energies (19 Wh g^−1^ at 1 A g^−1^) that were comparable with less-scalable thick electrodes previously reported.

## Figures and Tables

**Figure 1 materials-16-03439-f001:**
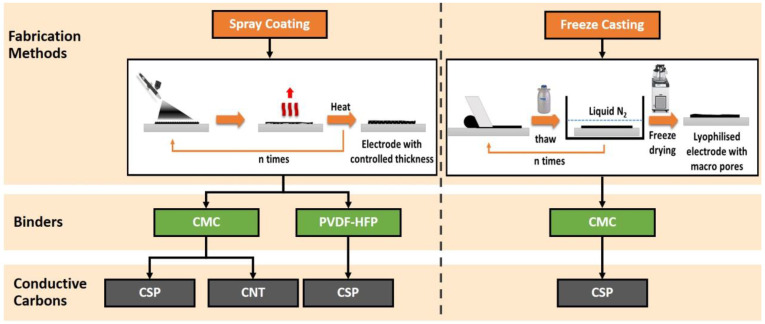
Electrode-making processes through spray coating and freeze casting.

**Figure 2 materials-16-03439-f002:**
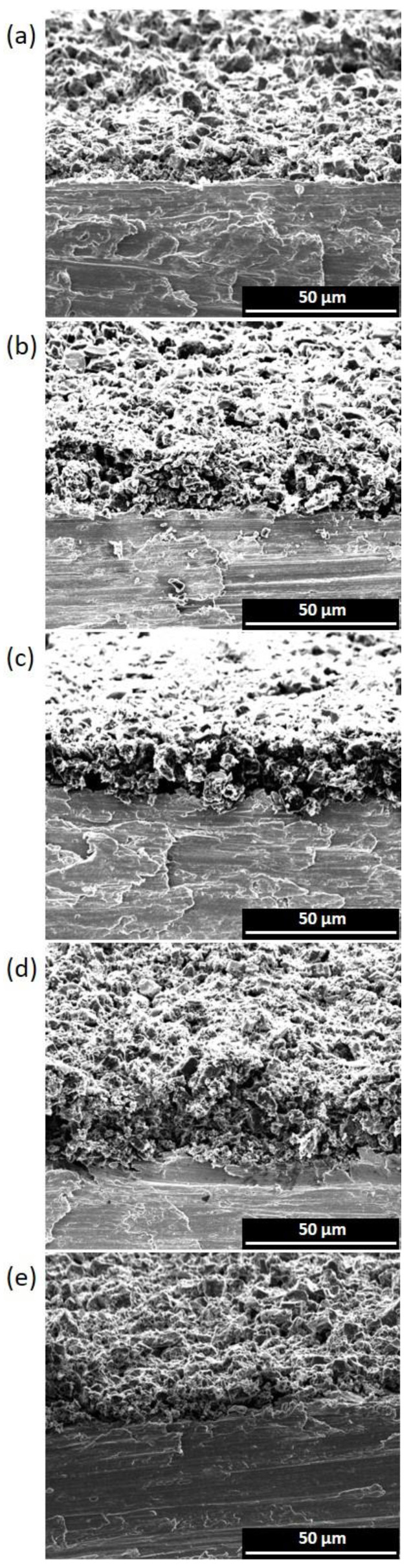
SEM images of thin electrodes (**a**) 1_CSP_5%CMC, (**b**) 1_CNT_5%CMC, (**c**) 1_CSP_10%PVDF-HFP, (**d**) 1_CSP_15%PVDF-HFP, (**e**) 2_LYO_5%CMC.

**Figure 3 materials-16-03439-f003:**
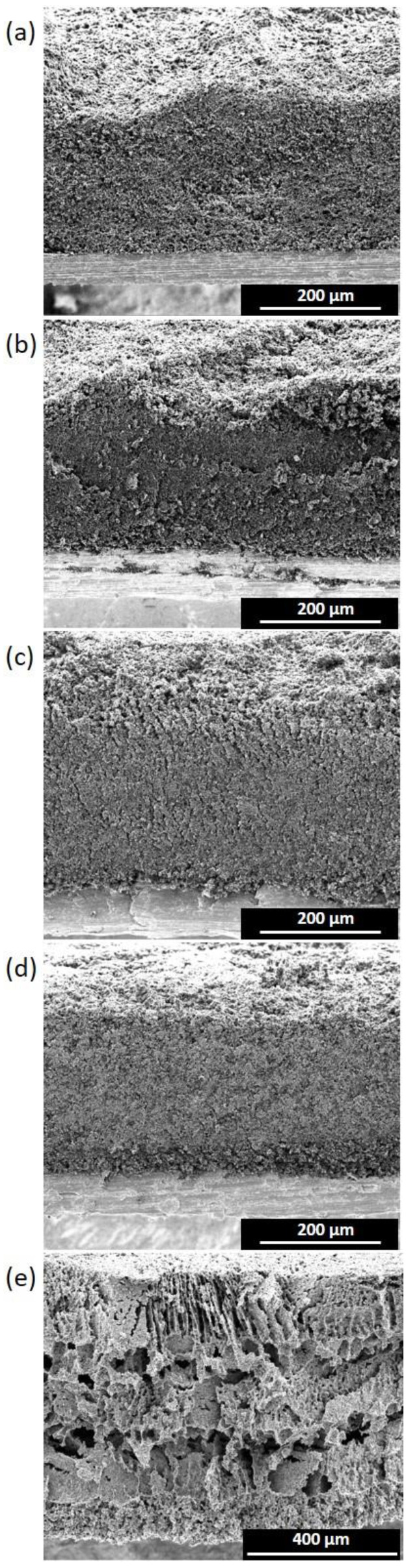
SEM images of thick electrodes (**a**) 4_CSP_5%CMC, (**b**) 4_CNT_5%CMC, (**c**) 4_CSP_10%PVDF-HFP, (**d**) 4_CSP_15%PVDF-HFP, (**e**) 3A_LYO_5%CMC.

**Figure 4 materials-16-03439-f004:**
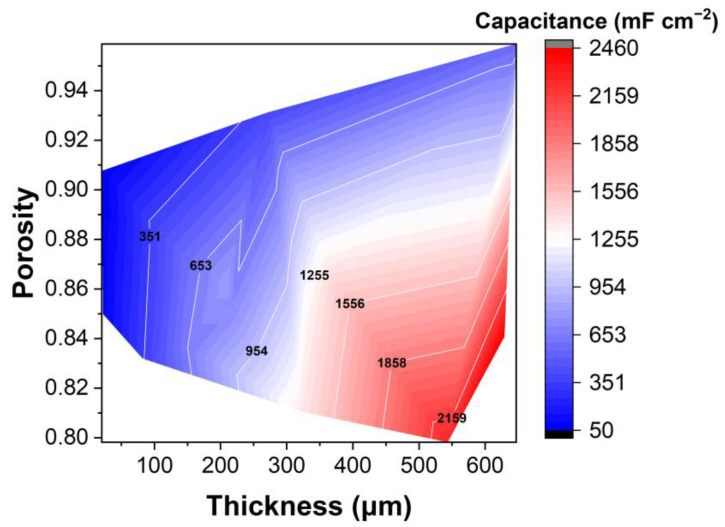
Dependence of electrode areal capacitance (at 10 mV s^−1^) on thickness and porosity (10 mV s^−1^).

**Figure 5 materials-16-03439-f005:**
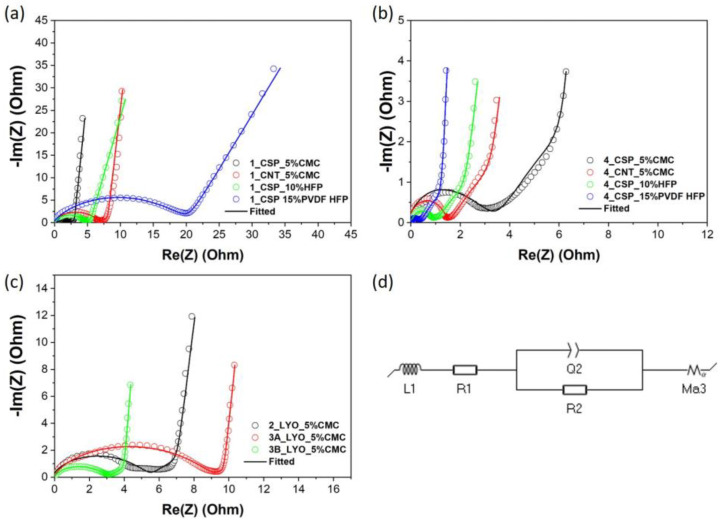
EIS fitting for (**a**) thin electrodes, (**b**) thick electrodes and (**c**) freeze-casted electrodes and (**d**) equivalent circuit.

**Figure 6 materials-16-03439-f006:**
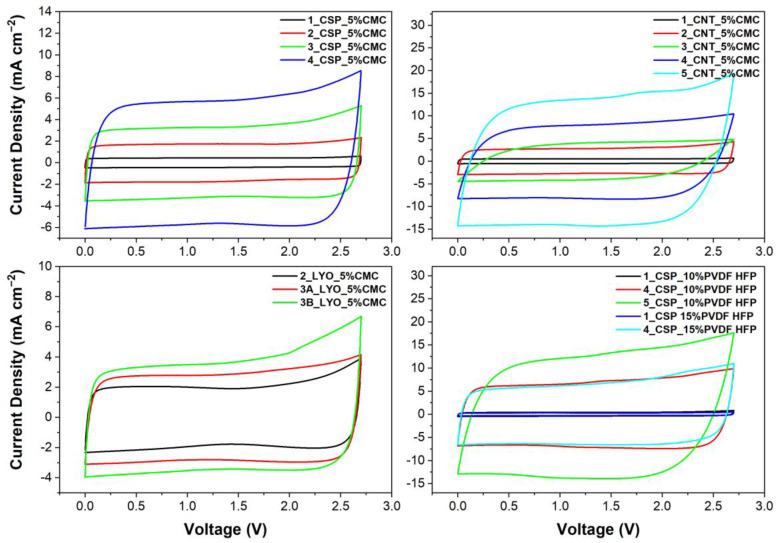
CPs of symmetric coin cells at 10 mV s^−1^ with 1M TEABF_4_ in ACN.

**Figure 7 materials-16-03439-f007:**
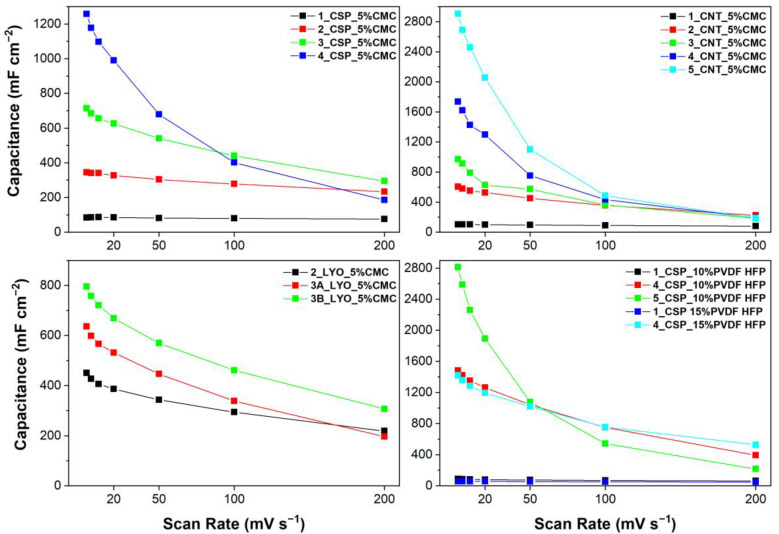
Areal capacitance at different scan rates for each type of electrode.

**Figure 8 materials-16-03439-f008:**
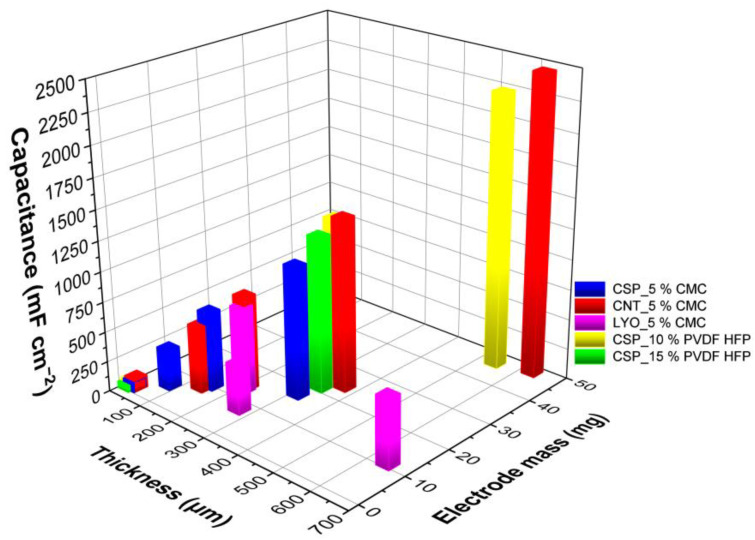
Capacitance (at 10 mV s^−1^) of various electrodes as a function of thickness and mass.

**Figure 9 materials-16-03439-f009:**
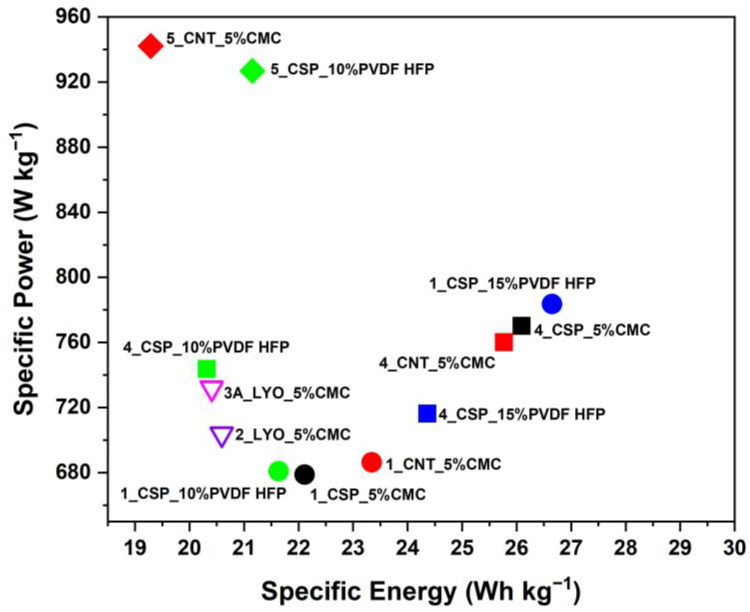
Ragone plot of electrodes with specific energy and specific power measured at 1 Ag^−1^. Here, circles represent spray-coated thin electrodes with mass loading ≤ 1 mg cm^−2^, squares represent spray-coated thick electrodes with average mass loading of 14 mg cm^−2^, diamond shapes indicate spray-coated thick electrodes with average mass loading of 31 mg cm^−2^, and inverted triangles depict freeze-casting electrodes.

**Figure 10 materials-16-03439-f010:**
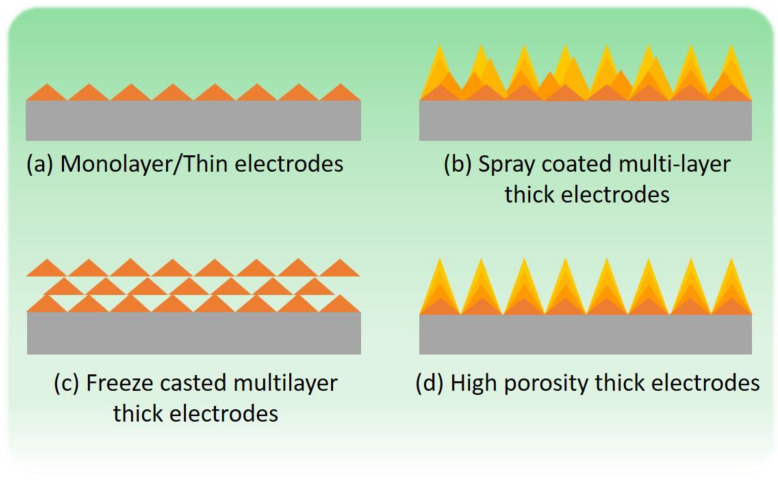
Schematic representation of multilayer electrode structures exemplifying different microstructures, each leading to different porosities and electrode–electrolyte active areas.

**Table 1 materials-16-03439-t001:** Specifications of electrodes.

	Electrode Mass (mg)	Thickness (µm)	Capacitance (10 mV s^−1^) (mF cm^−2^)	Capacitance (10 mV s^−1^) (F g^−1^)	Porosity	Tortuosity	Specific Energy (Wh kg^−1^)	Specific Power (W kg^−1^)
1_CSP_5%CMC	1.37	31	87	98	89%	2.5	22	679
1_CNT_5%CMC	1.70	31	103	94	89%	4.6	23	686
1_CSP_10%PVDF-HFP	1.43	23	83	89	85%	4.3	22	681
1_CSP 15%PVDF-HFP	0.84	21	58	107	91%	8.1	27	784
4_CSP_5%CMC	17.09	317	1098	99	87%	1.0	26	770
4_CNT_5%CMC	23.88	369	1428	92	86%	0.5	26	760
4_CSP_10%PVDF-HFP	25.74	322	1351	81	81%	0.4	20	744
4_CSP_15%PVDF-HFP	21.18	333	1285	93	86%	0.3	24	716
5_CNT_5%CMC	47.81	629	2459	79	84%	0.2	19	942
5_CSP_10%PVDF-HFP	46.32	543	2261	75	80%	0.1	21	927
2_LYO_5%CMC	7.54	271	407	83	93%	0.9	21	704
3A_LYO_5%CMC	10.78	647	567	81	96%	0.1	20	732

Electrode Mass (mg)

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
