# Peer review of "Nanostructured Thick Electrode Strategies toward Enhanced Electrode–Electrolyte Interfaces"

_materials, 2023, doi:10.3390/ma16093439_

Round 1

Reviewer 1 Report

This manuscript uses conventional, easily scalable methods of spray coating and freeze-casting to prepare thick electrodes. Authors compare different binders and conductive carbons to develop thick electrodes and analyse factors that determine the performance of such thick electrodes such as porosity and tortuosity. Moreover, the high electrochemical performances of spray-coated electrodes are exhibited. Thus, we can recommend its publication in Materials after minor revision. Here are my detailed comments:

1. The detailed information of all raw materials should be provided, such as manufacturer, purity and size.

2. Why are the gravimetric capacitances of thick electrodes higher than those of thin electrodes in Figure S1.

3. The resistances of 1_CSP_5%CMC are smaller than those of 1_CNT_5%CMC, while the resistances of 4_CSP_5%CMC are larger than those of 4_CNT_5%CMC. Please provide a reasonable explanation.

4. Some references on thick electrode are suggested to be read and cited (10.1016/j.jechem.2023.03.002 and 10.1002/advs.202204087).

There are some minor mistakes in manuscript, please check manuscript carefully. For example, “and75:10:15” should be “and 75:10:15”, “H2O” should be “H2O”.

The manuscript should be polished.

Reviewer 2 Report

This manuscript showed two preparation methods of thick electrodes with several kinds of conventional binders and conductive carbons. However, the language and the content are not well organized, especially in the Introduction and Conclusions parts. Moreover, the data discussions are insufficient and less of scientific analysis, such as the compact density, volumetric capacitance, et al. that are more concerned parameters for the thick electrodes. I don't think the novelty and scientific significance of this paper are very great and I think it not appropriate for Materials.

Extensive editing of English language required.

Reviewer 3 Report

The article "Nanostructured thick electrode strategies toward enhanced electrode-electrolyte interfaces" describes the role of electrode thickness in electrochemical performance. I am suggesting the following points to upgrade the manuscript to meet the journal's standards.

1.  In Fig. 2 and 3, it is better for the authors to provide SEM images of thin and thick electrodes with the same scale for a fair comparison.

2.      The authors explain their results based on the porosity of the electrodes. The authors can further support their results by including the BET surface area and pore size distribution curve.

3.      The authors can further explain the diffusion resistance based on the EIS. The authors can follow these reference articles: (https://doi.org/10.1016/j.carbon.2023.03.045, Fig. S10f, and Fig. 8d) and (https://doi.org/10.1016/j.nanoen.2019.103921, Fig. 6b).

4.      The conclusion of the manuscript should be short and highlight the critical finding of their work. The authors can move some of their discussion to the result and discussion section.

Moderate editing of the English language is required

Reviewer 4 Report

This work will provide additional/supporting information to the battery researchers and their field. But required revision to improve journal quality. 

In order to improve the quality of the manuscript authors need to verify the following comments,

1. Please concisely revise and improve the conclusion to better understand the readers.

2. Fig 2 & Fig 3 SEM images are not consistent and if possible  provide clear Surface SEM images with clear porosity( since it's discussed in the study )

3. All the figure's quality must be improved.

4. Why is the higher electrode mass of CNT_5% CMC over CSP_10 % PVDF HFP? Explain its significance.

Round 2

Reviewer 4 Report

All the given comments are explained and discussed in the revised version.